# Cystic fibrosis drug ivacaftor stimulates CFTR channels at picomolar concentrations

**László Csanády[1,2]\*, Beáta Töröcsik[1,2]**

[1]Department of Medical Biochemistry, Semmelweis University, Budapest, Hungary; [2]MTA-SE Ion Channel Research Group, Semmelweis University, Budapest, Hungary

**Abstract** The devastating inherited disease cystic fibrosis (CF) is caused by mutations of the Cystic Fibrosis Transmembrane Conductance Regulator (CFTR) anion channel. The recent approval of the CFTR potentiator drug ivacaftor (Vx-770) for the treatment of CF patients has marked the advent of causative CF therapy. Currently, thousands of patients are being treated with the drug, and its molecular mechanism of action is under intensive investigation. Here we determine the solubility profile and true stimulatory potency of Vx-770 towards wild-type (WT) and mutant human CFTR channels in cell-free patches of membrane. We find that its aqueous solubility is ~200 fold lower (~60 nanomolar), whereas the potency of its stimulatory effect is >100 fold higher, than reported, and is unexpectedly fully reversible. Strong, but greatly delayed, channel activation by picomolar Vx-770 identifies multiple sequential slow steps in the activation pathway. These findings provide solid guidelines for the design of *in vitro* studies using Vx-770.
DOI: https://doi.org/10.7554/eLife.46450.001

## Introduction

CFTR belongs to the family of ATP Binding Cassette (ABC) proteins (*Riordan et al., 1989*), and forms an anion selective channel which is activated by phosphorylation of its cytosolic regulatory (R) domain by cyclic AMP-dependent protein kinase (PKA) (*Berger et al., 1991*; *Tabcharani et al., 1991*). In phosphorylated CFTR channels opening and closing (gating) of the anion pore is coupled to conformational changes induced by ATP binding and hydrolysis at two cytosolic nucleotide binding domains (NBDs) (*Anderson et al., 1991*; *Li et al., 1996*). CFTR channels are present in the apical membrane of epithelial cells that line the lung, intestine, pancreatic duct, and sweat duct, and the regulated flow of anions, primarily chloride and bicarbonate, through CFTR is indispensable for the salt-water homeostasis of those epithelia. CF, the most common lethal inherited disease among Caucasians, is caused by CFTR mutations which have been classified based on their molecular consequences. Thus, some mutations diminish production (Class I and V), folding/trafficking (Class II), or stability (Class VI) of the CFTR protein, wheres others impair channel gating (Class III), or anion permeation through the open pore (Class IV) (*De Boeck and Amaral, 2016*). The most common CF mutation, deletion of phenylalanine 508 (ΔF508), is present in ~90% of patients and impairs both channel surface expression (*Cheng et al., 1990*) and open probability (*Miki et al., 2010*).

Pharmacotherapy of CF is currently focused on developing compounds that either enhance surface expression ('correctors') or stimulate channel gating ('potentiators') of mutant CFTR. The potentiator Vx-770 (*Van Goor et al., 2009*), identified by Vertex Pharmaceuticals using high-throughput screening, has proven successful and was approved by the FDA for the treatment of patients carrying G551D and other gating mutations (*Ramsey et al., 2011*). This breakthrough has demonstrated the feasibility of efficient causative CF therapy using small-molecule potentiators, at least for this subset of patients (<5% of all CF cases). Moreover, in a recent phase two clinical trial, patients

\*For correspondence:
csanady.laszlo@med.semmelweis-univ.hu

carrying the most common CF mutation ΔF508 also experienced significant clinical improvement from co-administration of Vx-770 with a combination of corrector drugs (*Boyle et al., 2014*; *Wainwright et al., 2015*; *Davies et al., 2018*).

Although treatment with Vx-770 (administered orally, either alone or in combination with corrector drugs) is accepted as the standard of care for the majority of CF patients in the US, many of the basic properties of the drug have remained unexplained or controversial. Thus, CFTR channel stimulation by Vx-770 in cell-free patches is reportedly irreversible (*Jih and Hwang, 2013*; *Lin et al., 2016*; *Yeh et al., 2015*; *Yeh et al., 2017*; *Wang et al., 2014*; *Wang et al., 2018*), maximal stimulation of WT CFTR channels shows large variability (ranging from ~1.1 fold [*Cui and McCarty, 2015*] to ~3 fold [*Wang et al., 2018*]), and in some studies its acute application caused (*Cholon et al., 2014*) – or accentuated (*Wang et al., 2014*) – CFTR channel inactivation. Vx-770 is routinely used at concentrations up to 10 μM (*Cui and McCarty, 2015*; *Langron et al., 2018*; *Liu and Dawson, 2014*; *Van Goor et al., 2009*; *Van Goor et al., 2014*; *Yu et al., 2012*; *Wang et al., 2014*; *Wang et al., 2018*), but data on its aqueous solubility have not yet been published.

## Results

### Aqueous solubility of Vx-770 is two orders of magnitude lower than generally assumed

We reasoned that some of the reported controversies on Vx-770 effects might be explained by the use of highly supersaturated concentrations of the drug in all published studies, and set out to determine its true solubility profile. Starting from crystalline Vx-770 (Selleck Chemicals), solubilities (S) of the drug in various organic solvents could be readily determined by adding incremental small volumes of solvent until all crystals had been dissolved (*Table 1*). We found Vx-770 extremely soluble in anhydrous dimethyl sulfoxide (DMSO) (S ~ 0.77 M), and reasonably soluble in 1-octanol (S ~ 3.9 mM) and ethanol (S ~ 3.1 mM). However, even the smallest amounts that we could weigh out (~0.1 mg) failed to dissolve in 1 liter of aqueous saline (pH = 7.1), as evident from clearly visible crystals even after vigorous shaking for 24 hr, forcing us to take a different approach. To generate an aqueous solution exactly saturated with Vx-770, a small aliquot of crystals was added to an aqueous saline (pH = 7.1) and, after vigorous shaking for 24 hr, visible crystals were removed by repeated filtering and microcrystals sedimented by centrifugation (Materials and methods). The resulting supernatant (our '1x saturated' stock solution) was then used both for determination of Vx-770 solubility (*Figure 1b–c*), and for functional experiments (*Figures 2–5*). Vx-770 dissolved in 1-octanol at 1–4 μM showed clearly measurable light absorption with a peak at 311 nm (*Figure 1b*, *blue spectra*), yielding a calibration line in the micromolar range (*Figure 1c*, *blue dots* and *fitted straight line*). For our 1x saturated aqueous stock solution the absorbance spectrum was flat against the background saline, but when we extracted it with 1-octanol at a volume ratio of 20:1, the extract showed clear absorbtion (*Figure 1b*, *red spectrum*, '1st extract'), corresponding to a Vx-770 concentration of ~1.2 μM (*Figure 2b*). Repeated extraction of the aqueous stock with 1-octanol yielded a flat spectrum (*Figure 2a*, *green spectrum*, '2nd extract'), confirming that all Vx-770 had been efficiently extracted

**Table 1.** Solubilities of Vx-770 in various solvents at 25°C.
Solubilities were determined as described in detail in Materials and methods. *Aqueous saline with composition described in Materials and methods.

| Solvent | S (mol/L) |
|---|---|
| water (pH=7.1)* | $6.2 \cdot 10^{-8}$ |
| pentane | $1.8 \cdot 10^{-5}$ |
| octane | $2.6 \cdot 10^{-3}$ |
| ethanol | $3.1 \cdot 10^{-3}$ |
| 1-octanol | $3.9 \cdot 10^{-3}$ |
| DMSO | $7.7 \cdot 10^{-1}$ |

DOI: https://doi.org/10.7554/eLife.46450.002

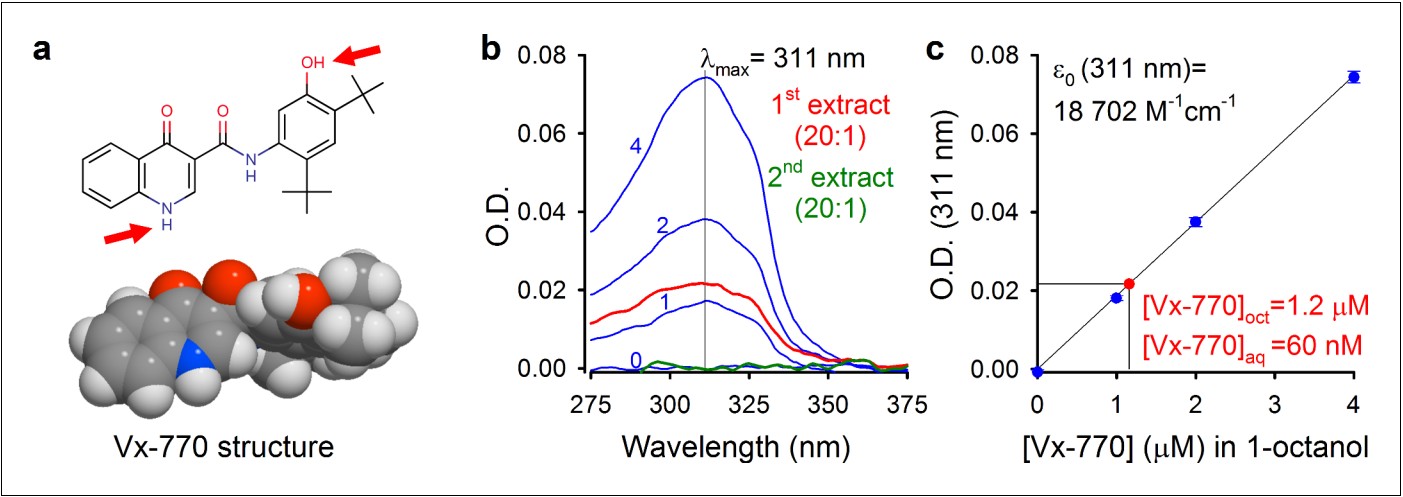

**Figure 1.** Determination of aqueous solubility for Vx-770. (a) Structural formula of Vx-770 and 3D surface rendering by Molinspiration (https://www.molinspiration.com/cgi-bin/properties). *Red arrows* identify potentially deprotonatable groups. (b) Absorption (O.D.) spectra for 0, 1, 2, and 4 µM Vx-770 dissolved in 1-octanol (*blue spectra*), and for two sequential 1-octanol extracts (20:1 volume ratio) of an aqueous saline saturated with Vx-770 (*red and green spectra*). (c) Calibration curve of O.D. at 311 nm for Vx-770 in 1-octanol (*blue dots* (mean ± S.E.M. from 3 measurements) and *linear regression line*), and quantification of [Vx-770] in the first 1-octanol extract (*red dot*).
DOI: https://doi.org/10.7554/eLife.46450.003

in the first round. Considering that Vx-770 was concentrated 20-fold during the extraction procedure, we conclude that the true aqueous solubility of Vx-770 at 25°C is only ~60 nM ($S_{aq}$(25°C, pH = 7.1)=62 ± 2 nM, n = 4), almost 200-fold lower than generally believed, and severalfold lower even than its reported $EC_{50}$ values for potentiation of various CFTR mutants (160–600 nM) (*Van Goor et al., 2014*).

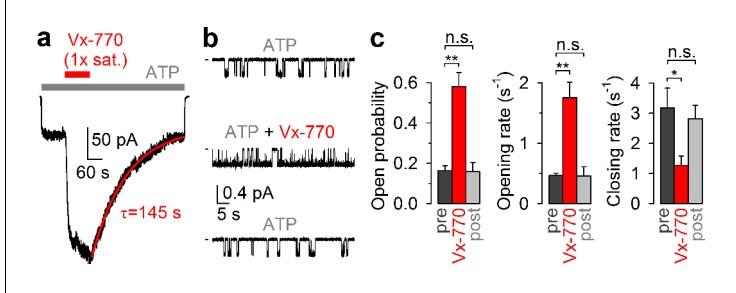

**Figure 2.** CFTR current stimulation by Vx-770 is fully reversible. (a) Macroscopic WT CFTR channel current, elicited in an inside-out patch by exposure to 2 mM ATP (*gray bar*), is enhanced ~4-fold by application of a 1x saturated (~62 nM) Vx-770 solution (*red bar*), and relaxes back to its pre-drug level following drug removal. *Red line* is a fitted exponential with time constant (τ) indicated. Channels had been pre-phosphorylated by an ~1-min exposure to 300 nM PKA catalytic subunit, membrane potential is -40 mV. (b) 1-min segments of recording from a single pre-phosphorylated WT CFTR channel gating in 2 mM ATP before drug exposure (*top*), in the presence of 0.05x saturated (~3 nM) Vx-770 (*center; segment starts ~4 min after drug addition*), and long after Vx-770 removal (*bottom, segment starts ~9 min after drug removal*). (c) Open probabilities (*left*), opening rates (*center*), and closing rates (*right*) of single pre-phosphorylated WT CFTR channels gating in 2 mM ATP, before (*dark gray bars*), during (*red bars*), and after (*light gray bars*) exposure to 0.05x saturated (~3 nM) Vx-770. Bars show mean ± S.E.M. from 9 experiments.
DOI: https://doi.org/10.7554/eLife.46450.004

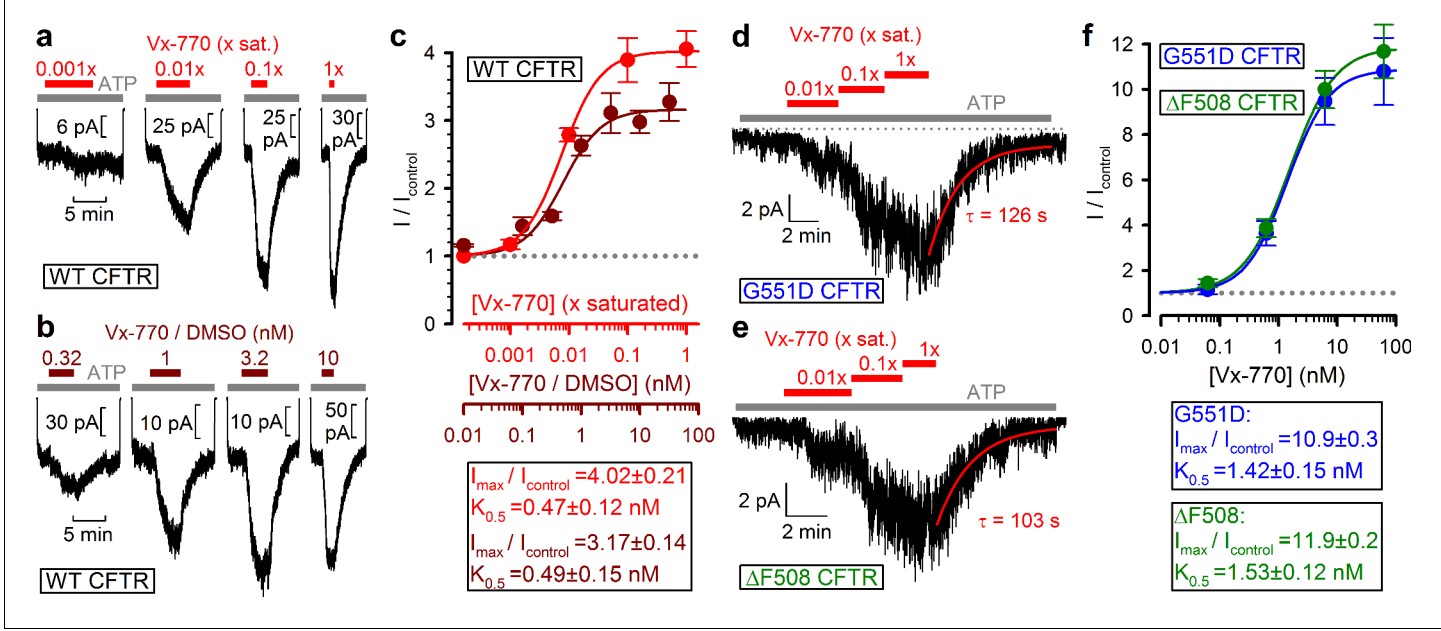

**Figure 3.** Vx-770 stimulates CFTR currents already at subnanomolar concentrations. (**a-b**) Macroscopic WT CFTR currents elicited by 2 mM ATP are reversibly stimulated by exposure (*red* and *brown bars*) to indicated concentrations of Vx-770, diluted either from a 1x saturated aqueous stock (**a**) or from a 10-mM stock dissolved in DMSO (**b**). (**c**) Fractional stimulation of WT CFTR currents by Vx-770 diluted from a 1x saturated aqueous (*red symbols and abscissa*), or a DMSO-based (*brown symbols and abscissa*), stock. Abscissae are aligned based on the aqueous solubility of Vx-770 (~62 nM). (**d–e**) Quasi-macroscopic currents of prephosphorylated G551D (**d**) and ΔF508 (**e**) CFTR channels in 2 mM ATP are reversibly stimulated by exposure (*red bars*) to indicated concentrations of Vx-770, diluted from a 1x saturated aqueous stock. Deactivation time courses following drug removal are fitted with single exponentials (*red lines*). (**f**) Fractional stimulation of G551D (*blue symbols*) and ΔF508 (*green symbols*) CFTR currents by Vx-770 diluted from a 1x saturated aqueous stock; abscissa has been calibrated. *Symbols* in (**c**) and (**f**) show mean ± S.E.M. from 3-13 experiments, *solid curves* are fits to an adapted Hill equation (Materials and methods) with parameters plotted. Hill coefficients were 1.31±0.65 and 1.32±0.47 for WT in Vx-770 or Vx-770/DMSO, 1.12±0.07 for ΔF508, and 1.22±0.10 for G551D CFTR.

DOI: https://doi.org/10.7554/eLife.46450.005

## CFTR Channel Stimulation by Vx-770 is fully reversible

Because in all published *in vitro* studies Vx-770 was applied at concentrations ranging from 100 nM to 10 μM (*Cui and McCarty, 2015*; *Yeh et al., 2015*; *Yeh et al., 2017*; *Langron et al., 2018*; *Liu and Dawson, 2014*; *Van Goor et al., 2009*; *Van Goor et al., 2014*; *Yu et al., 2012*; *Wang et al., 2014*; *Wang et al., 2018*; *DeStefano et al., 2018*; *Jih and Hwang, 2013*; *Lin et al., 2016*; *Kopeikin et al., 2014*), our present solubility estimate indicates that Vx-770 effects have so far been characterized only at supersaturated concentrations. We reasoned that this circumstance might explain some of the puzzling features of the drug, such as CFTR channel inactivation observed in some cases (*Cholon et al., 2014*), as well as apparent irreversibility of drug effects (*Jih and Hwang, 2013*; *Lin et al., 2016*; *Yeh et al., 2015*; *Yeh et al., 2017*; *Wang et al., 2014*; *Wang et al., 2018*). To investigate stability and reversibility of Vx-770 effects in a concentration range that can be achieved in the human body during oral drug administration, we tested the effects of our 1x saturated Vx-770 solution and its dilutions in inside-out patches excised from *Xenopus laevis* oocytes expressing WT or mutant human CFTR channels.

Indeed, in inside-out patches (*Figure 2a*), application of our 1x saturated Vx-770 solution (*red bar*) to prephosphorylated WT CFTR channels opened by 2 mM ATP (*gray bar*) caused rapid robust (~4 fold) current stimulation which lasted as long as the drug was applied. However, upon drug removal the current readily declined back to its pre-drug value (time constant τ ~2 min, *Figure 2a*, *red fit line*), demonstrating full reversibility of Vx-770 effects. In single-channel recordings (*Figure 2b*), even a 20-fold dilution of our 1x saturated Vx-770 stock stimulated open probability of WT CFTR to a comparable level, by ~4 fold (*Figure 2c*, *left*). Stimulation of open probability was due to acceleration of channel opening and slowing of channel closure (*Figure 2b*, *top* and *center trace*;

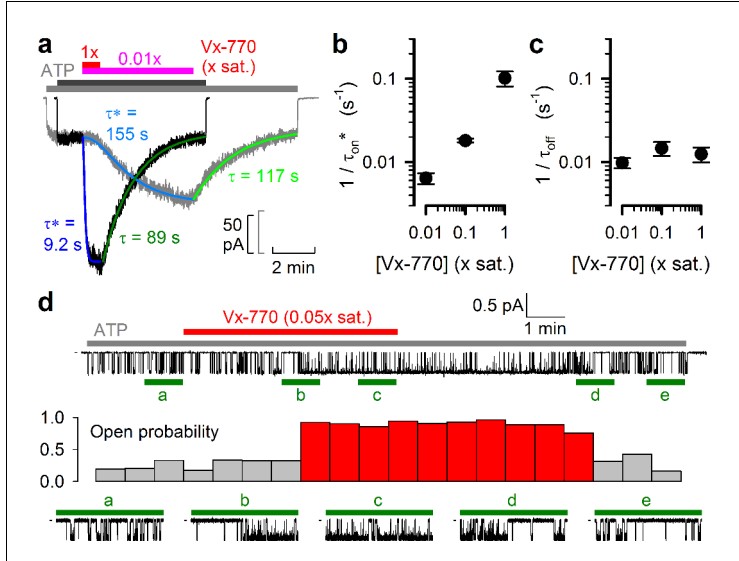

**Figure 4.** Delayed activation by low nanomolar Vx-770 suggests multiple sequential slow steps in activation process. (**a**) Stimulation of macroscopic WT CFTR currents in 2 mM ATP by exposure to a 1x (*black trace; dark gray and red bar*) or an 0.01x (*gray trace; light gray and purple bar*) saturated aqueous solution of Vx-770. Current amplitudes have been rescaled by their pre-drug values, and superimposed traces are shown synchronized to the time point of drug addition. Sigmoidal activation time courses are fitted (*blue lines*) by a sequential three-step mechanism, and apparent activation time constants ($\tau^*$, see Materials and methods) are plotted. Deactivation time courses are fitted (*green lines*) by single exponentials with time constants ($\tau$) indicated. (**b–c**) Overall rates of current stimulation ($1/\tau_{on}^*$; b) and rates of deactivation ($1/\tau_{off}$; c) in response to addition and removal, respectively, of indicated concentrations of Vx-770. Mean ± S.E.M. from 5-7 experiments. (**d**) (*Top*) Continuous ~16-min recording from a single prephosphorylated WT CFTR channel gating in 2 mM ATP (*gray bar*) and exposed for ~6 min to 0.05x saturated (~3 nM) Vx-770 (*red bar*). (*Center*) Open probability calculated over sequential 46-s intervals for the recording shown on top. Note sudden switch from low- (*gray bars*) to high-activity gating (*red bars*) ~3 min after initiation of drug exposure, and a similarly sudden switch back to low-activity gating ~5 min after drug removal. (*Bottom*) 1-min segments marked by green bars in the recording on top are shown at an expanded time scale.

DOI: https://doi.org/10.7554/eLife.46450.006

*Figure 2c*, *red* vs. *dark gray bars*), consistent with previous studies in which the drug had been applied at supersaturating concentrations (*Kopeikin et al., 2014*; *Jih and Hwang, 2013*). However, in contrast to those earlier studies, the observed effects on channel gating were fully reversible following washout of the drug (*Figure 2b*, *bottom trace*; *Figure 2c*, *light gray bars*).

## Vx-770 stimulates WT and mutant CFTR channels already at subnanomolar concentrations

To estimate the true stimulatory potency of the drug, a series of dilutions of our 1x saturated Vx-770 stock was first tested on WT CFTR channels (*Figure 3a*, *red bars*). We observed robust stimulation of WT CFTR currents by even a 100-fold dilution of our 1x saturated Vx-770 stock (*Figure 3a*, *second trace from left*), suggesting a subnanomolar apparent affinity for the drug. Indeed, a dose response curve (*Figure 3c*, *red symbols*), obtained by assuming [Vx-770]=62 nM in our 1x saturated stock (cf., *Table 1*), yielded a $K_{0.5}$ of 0.47 ± 0.12 nM (*Figure 3c*, *red fit line*). As a control, a dose response relationship obtained by exposure of WT CFTR channels to known concentrations of Vx-770 (*Figure 3b*, *brown bars*), obtained by sequential dilutions of a commercially available 10 mM DMSO-based Vx-770 stock (Selleck Chemicals), yielded a $K_{0.5}$ for current stimulation of 0.49 ± 0.15 nM (*Figure 3c*, *brown symbols* and *fit line*), confirming our estimate of Vx-770 aqueous solubility. Thus, the apparent $K_{0.5}$ of Vx-770 for stimulation of WT CFTR is ~600 fold lower than previously reported (~300 nM [*Van Goor et al., 2014*; *Langron et al., 2018*]).

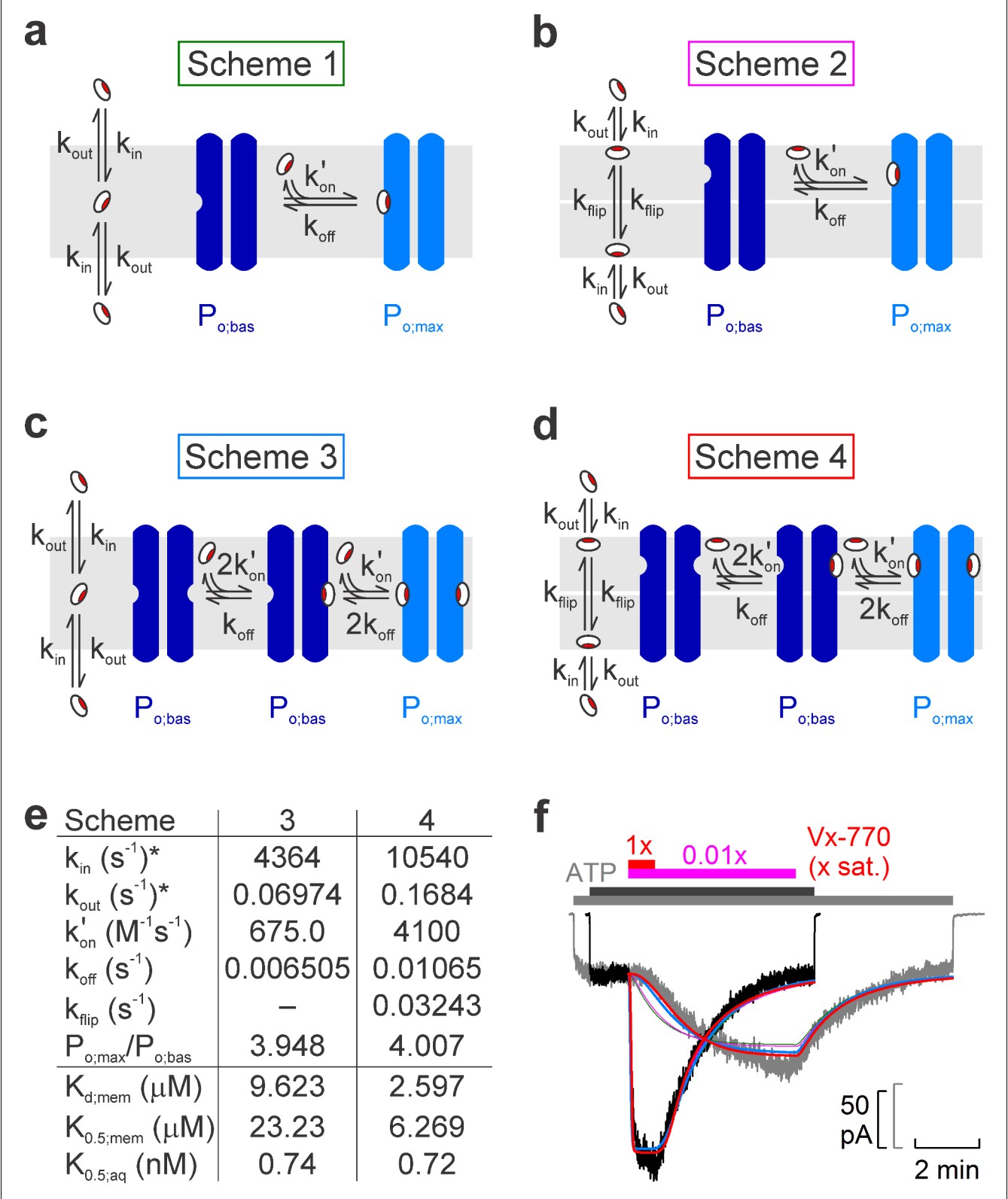

**Figure 5.** Model fitting of activation/deactivation time courses in response to cytosolic addition/removal of Vx-770. (**a-d**) Cartoon representation of schemes with one (**a, b**) or two (**c, d**) drug binding sites, and one (**a, c**) or two (**b, d**) membrane compartments (cytoplasmic, bottom; extracellular, top). (**e**) Fit parameters obtained from global fits of Schemes 3 (*left column*) and 4 (*right column*) to the data set shown in f. *Parameters $k_{in}$ and $k_{out}$ were constrained through $k_{in}/k_{out}$ = $D_{oct/wat(pH=7.1)}$ = 62580. Parameters $K_{d;mem}$, $K_{0.5;mem}$, and $K_{0.5;aq}$ were calculated from the fit parameters as described in
*Figure 5 continued on next page*

*Figure 5 continued*

Materials and methods. (f) Pair of normalized macroscopic WT CFTR current activation/deactivation time courses recorded in response to cytosolic addition/removal of 0.01x saturated (*gray trace*) and 1x saturated (*black trace*) concentrations of Vx-770 (replotted from *Figure 4a*), and ensemble fits by the models in a–d (*colored lines; green*, Scheme 1, *pink*, Scheme 2, *blue*, Scheme 3, *red*, Scheme 4). See also *Figure 5—figure supplement 1*.
DOI: https://doi.org/10.7554/eLife.46450.007
The following figure supplement is available for figure 5:

**Figure supplement 1.** Scheme 2 adequately fits individual on- and off- time courses but not the global data set.
DOI: https://doi.org/10.7554/eLife.46450.008

When tested on patches containing G551D (*Figure 3d*) or ΔF508 (*Figure 3e*) CFTR channels, robust current stimulation was also observed already at subnanomolar concentrations. For both mutants maximal current stimulation was 11–12-fold with a $K_{0.5}$ of ~1.5 nM (*Figure 3f*, *blue and green symbols* and *fit lines*), and was fully reversible with a time constant of ~2 min (*Figure 3d–e*, *fitted lines*). Thus, the true potency of Vx-770 for G551D CFTR is ~200 fold higher, and for ΔF508 CFTR 20–300-fold higher, than previously reported (*Van Goor et al., 2009*; *Yu et al., 2012*; *Van Goor et al., 2014*; *Langron et al., 2017*).

## Delayed channel activation by low concentrations of Vx-770 suggests multiple slow steps in the activation pathway

The kinetics of current deactivation following Vx-770 wash-off was well fitted assuming a single rate-limiting step (*Figure 4a*, *green single-exponential fit lines* and *time constants* τ). In contrast, the activation time course upon drug exposure was clearly sigmoidal, suggesting the presence of multiple sequential slow steps in the activation pathway. Indeed, at least three sequential concentration-dependent slow steps were required to obtain an adequate fit (*Figure 4a*, *blue fit lines*). In particular, at subnanomolar/low nanomolar aqueous drug concentrations current stimulation started off with a marked delay of up to ~1 min (*Figure 4a*, *gray trace*), consistent with slow drug accumulation in the membrane being required for CFTR potentiation (*Jih and Hwang, 2013*). As expected, deactivation rate ($1/\tau_{off}$) was independent of drug concentration (*Figure 4c*), whereas the overall rate of activation ($1/\tau_{on}$*, see Materials and methods) became faster at higher concentrations (*Figure 4b*). At a single-channel level both application and removal of low concentrations of Vx-770 (*Figure 4d*, *top*, *red bar*) caused abrupt changes in channel open probability, but with a marked delay (*Figure 4d*, stability plot, *red* vs. *gray bars*; bottom, *expanded current segments*).

## Time courses of channel activation and deactivation by Vx-770 are well fitted by a model with two intramembrane drug binding sites

We next attempted to interpret the observed complex kinetics of the drug effects in terms of various molecular models (*Figure 5a–d*), by performing global fits of each model (*Figure 5f*, *colored lines*) to the ensemble of four experimentally observed current time courses: activation and deactivation upon addition and removal of 0.01x saturated (*Figure 5f*, *gray trace*) and 1x saturated (*Figure 5f*, *black trace*) Vx-770. To reduce the number of free parameters in the considered models, the equilibrium ratio of membrane-dissolved and aqueous concentrations of Vx-770 (i.e., the ratio $k_{in}/k_{out}$, *Figure 5a–d*) was fixed to its measured octanol/water distribution coefficient ($D_{oct/wat(pH=7.1)}$=62580, logD = 4.8, cf., *Table 1*), a well-established approximator of membrane/water distribution for a large number of diverse drugs with logD values ranging between 1 and 5.5 (*Gobas et al., 1988*).

One slow step involved in both activation and deactivation might be drug equilibration between the membrane and the aqueous phase (described by rate constants $k_{in}$ and $k_{out}$), and/or binding/unbinding between membrane-dissolved drug and the channel protein (described by rate constants $k_{on}'$ and $k_{off}$). However, a simple two-step mechanism (Scheme 1, *Figure 5a*) was clearly insufficient to account for the marked delay in channel activation observed at low drug concentrations (*Figure 5f*, *green fit lines*).

Given that in inside-out patch-clamp experiments Vx-770 enters the membrane only from the cytosolic side during drug application, whereas it can leave the membrane on both sides following drug wash-off, one additional slow step specific to the activation process could be slow flipping of the drug into the outer membrane leaflet, if the binding site on the CFTR protein resided on the

extracellular half of its transmembrane region (Scheme 2, *Figure 5b*, flip-flop rates are assumed symmetrical ($k_{flip}$)). However, whereas such a two-compartment membrane model provides sufficient flexibility for perfectly fitting any individual, highly sigmoidal, on- time course (*Figure 5—figure supplement 1*, *solid pink fit line*), or even for reasonably describing an entire on-off current time course for any given drug concentration (*Figure 5—figure supplement 1*, *solid red and blue fit lines*), the ensemble fit of the entire data set by Scheme 2 (*Figure 5f*, *pink fit lines*) was barely improved relative to the fit by Scheme 1 (*Figure 5f*, *green fit lines*), despite the introduction of an additional free parameter ($k_{flip}$).

An alternative mechanism that could explain delayed activation by the drug is the presence of multiple drug binding sites, all of which must be occupied to switch the channel to a high open probability state. Of note, a recent study suggested two potential binding sites for Vx-770 in the transmembrane domains of CFTR (*Yeh et al., 2019*). We therefore evaluated a simplistic model which postulates independent drug binding to two sites, with identical affinities, but a substantial change in open probability (from $P_{o;bas}$ to $P_{o;max}$) only for diliganded channels (Scheme 3, *Figure 5c*). Interestingly, although it contains no additional free parameters relative to Scheme 1, Scheme 3 provided a dramatically improved ensemble fit, yielding a single set of rate constants (*Figure 5e*, *left*) that can reasonably account for the current activation and deactivation time courses observed in response to addition and removal of Vx-770, over a 100-fold concentration range (*Figure 5f*, *blue fit lines*).

Finally, assuming a two-compartment membrane and two drug-binding sites, both accessible from the outer leaflet (Scheme 4, *Figure 5d*), afforded a further small improvement (*Figure 5f*, *red fit lines*) relative to Scheme 3, albeit at the expense of an extra free parameter (*Figure 5e*, *right*).

## pH- and temperature dependence of Vx-770 solubility

The pH of our standard bath solution (7.1) resembles intracellular pH, but extracellular pH in the human body is typically higher (~7.4). This transmembrane pH gradient might influence the distribution of Vx-770 in the body, as the drug contains two potentially deprotonatable groups (*Figure 1a*, *red arrows*), and the negative charge acquired upon deprotonation is expected to greatly enhance its aqueous solubility. Because the computationally (Chemaxon) predicted strongest acidic $pK_a$ value varies between 6.57 and 8.66, depending on the method of prediction, we measured Vx-770 solubility in our standard bath solution after adjusting its pH to 7.4 (see Materials and methods). From three independent measurements we obtained an estimate of $S_{aq}$(25°C, pH = 7.4) = 50 ± 1 nM (mean ± S.E.M.), which is not larger than that obtained for pH = 7.1, the small difference likely reflecting experimental limitations. This finding suggests that the strongest acidic $pK_a$ value of the drug is substantially higher than 7.4, and so the uncharged, fully protonated form remains the dominant microspecies in the entire physiological pH range. The corollary is that the aqueous solubility of the drug is similar in the cytosol and the extracellular space.

As a further parameter of clinical interest, we also determined the aqueous solubility of Vx-770 at body temperature (pH = 7.1). Consistent with reports for other hydrophobic drugs (*Baena et al., 2004*; *Garzón and Martínez, 2004*; *Mota et al., 2009*), Vx-770 aqueous solubility at 37°C was ~2 fold higher compared to that at 25°C, three independent measurements yielding an estimate of $S_{aq}$(37°C, pH = 7.1) = 138 ± 1 nM (mean ± S.E.M.). The standard enthalpy of solution, estimated from the solubilities at 37°C vs. 25°C, is $\Delta H^{\circ}_{sol}$ = +51.2 kJ/mol, and the standard free energy of solution at 25°C is $\Delta G^{\circ}_{sol}$ = +41.1 kJ/mol (Materials and methods). Thus, the thermodynamic explanation for the extremely low aqueous solubility of Vx-770 is that the entropy increase ($T\Delta S^{\circ}_{sol}$ = +10.1 kJ/mol at 25°C) associated with the solution process is too small to compensate for the large increase in enthalpy, likely caused by the disruption of hydrogen bonds between water molecules.

## Discussion

The discovery of the CFTR potentiator Vx-770 has caused a major change in the therapy of CF patients, allowing to address for the first time the root cause of the disease. To date, thousands of patients are taking the drug on a daily basis, which warrants that studying its mechanisms will remain in the center of research efforts for years to come. As for all drugs approved for clinical use, extensive clinical studies have established its pharmacokinetic profile and safety features, and quantified its clinical effects. On the other hand, many of its basic physicochemical properties, as well as its

precise molecular mechanism of action, remain to be established. Vx-770 shows good oral bioavailability, and peak plasma concentrations of ~4 µM, and >10 µM, respectively, have been measured in patients receiving twice daily 150 mg (therapeutic) or 450 mg (supratherapeutic) doses (FDA report by Vertex Pharmaceuticals Incorporated, https://www.accessdata.fda.gov/drugsatfda_docs/nda/2012/203188Orig1s000OtherRedt.pdf). In the plasma, most of the drug is bound to plasma proteins, which are important for its pharmacokinetics, as they tremendously increase the drug transport capacity of the blood and therefore oral bioavailability, and act as a drug buffer system that replenishes the free drug pool during the time course of its elimination from the body. On the other hand, cell membranes are at equilibrium with free dissolved Vx-770, not with the protein-bound drug. Therefore, in a living organism the target cell membranes will accumulate Vx-770 only to an extent that is at equilibrium with free dissolved Vx-770 (in the extracellular fluid and/or in the cytosol). In the present study, we have established an upper limit for the latter parameter, which thus determines the maximal possible drug concentration achievable in target cell membranes. We have further shown that Vx-770 solubility is independent of pH in the physiological pH range, which implicitly indicates that its smallest acidic $pK_a$ value is far higher than 7.4, and that the dominant microspecies is therefore the fully protonated uncharged form. Consequently, the distribution of the drug between the cytosol and extracellular space is unaffected by the transmembrane pH gradient and the membrane potential.

Because in all published *in vitro* studies Vx-770 was applied at concentrations ranging from 100 nM to 10 µM (*Cui and McCarty, 2015*; *Yeh et al., 2015*; *Yeh et al., 2017*; *Langron et al., 2018*; *Liu and Dawson, 2014*; *Van Goor et al., 2009*; *Van Goor et al., 2014*; *Yu et al., 2012*; *Wang et al., 2014*; *Wang et al., 2018*; *DeStefano et al., 2018*; *Jih and Hwang, 2013*; *Lin et al., 2016*; *Kopeikin et al., 2014*), our present solubility estimate (50–62 nM at 25°C, 138 nM at 37°C) indicates that Vx-770 effects have so far been characterized only at highly (up to ~200 fold) supersaturated concentrations, that also exceed (by up to ~100 fold) the highest free drug concentrations that can be ever attained at 37°C, that is in the human body using oral drug administration. This circumstance readily explains some of the reported puzzling features of the drug. For instance, when superfusing a patch with a supersaturated aqueous solution of Vx-770, the drug will accumulate at supersaturated concentrations even in the membrane, likely resulting in the formation, and time-dependent growth, of crystalline precipitates within the bilayer. Such precipitates might have contributed to CFTR channel inactivation observed in some studies (*Cholon et al., 2014*). Moreover, following removal of Vx-770 from the aqueous perfusate, the membrane-dissolved drug pool might be replenished from such intra-membrane crystalline precipitates as long as the crystals last (which will depend on the degree of supersaturation of the applied aqueous test solution, and on exposure time), explaining the reported irreversibility of drug effects (*Jih and Hwang, 2013*; *Lin et al., 2016*; *Yeh et al., 2015*; *Yeh et al., 2017*; *Wang et al., 2014*; *Wang et al., 2018*). Indeed, we show here that the potentiating effect of Vx-770 on CFTR is fully reversible (*Figures 2–3*), as long as supersaturating drug concentrations are avoided.

Based on our data (*Figure 3c,f*), the apparent affinity of Vx-770 for WT CFTR is 600-fold, for G551D CFTR ~200 fold, and for ΔF508 CFTR 20–300-fold higher than previously reported. One likely explanation for the discrepancy between previous estimates of Vx-770 potencies and those obtained here is the use of static recording chambers in the previous studies as opposed to the continuous superfusion employed here. The octanol/water distribution coefficient of Vx-770 is ~60000 (cf., *Table 1*). Thus, if cellular membranes constitute as little as ~0.17% of the total assay volume, then ~99% of the drug will accumulate in those membranes and its free aqueous concentration will remain only ~1% of the total. Furthermore, when applied at supersaturated concentrations, large fractions of the drug will precipitate onto the walls of the recording chamber.

Precise molecular interpretation of the complex kinetic steps involved in drug-activation (*Figure 4a*) will clearly require much further work, including definitive localization of the drug binding site(s). However, a simple kinetic model which postulates two independent and similar-affinity binding sites, with potentiation requiring simultaneous binding of two drug molecules (Scheme 3, *Figure 5c*), afforded a remarkably good description of current activation/deactivation time courses over a broad range of applied drug concentrations (*Figure 5f*, *blue fit lines*). The reasonable fit of an ensemble of four highly non-linear curves by a single set of only four free parameters ($k_{out}$, $k_{on}'$, $k_{off}$, and $P_{o;max}/P_{o;bas}$; with $k_{in}$ constrained to $k_{in} = k_{out} \cdot D_{oct/wat(pH=7.1)}$) strongly argues in favor of this model. Recently, *in silico* docking to high-resolution CFTR structures of the

potentiator GLPG-1837, a drug known to compete with Vx-770 for CFTR activation (*Yeh et al., 2017*), identified potential drug binding sites in CFTR's transmembrane domains (*Yeh et al., 2019*). Interestingly, mutations introduced into either of two putative sites strongly impaired potentiation by both GLPG-1837 and Vx-770 (*Yeh et al., 2019*). However, because dose response curves for GLPG-1837 activation are fitted with Hill coefficients close to unity, the authors were uncertain how to interpret those findings. They suggested that either both sites are true drug binding sites, or only one site is a true binding site, while mutations at the other site allosterically affect the structure of the true site. Of note, Scheme 3 predicts an apparent Hill slope of only ~1.17 (see Materials and methods) which is consistent with the values obtained both for GLPG-1837 by Yeh and colleagues, and for Vx-770 in the present study (*Figure 3*). We therefore consider Scheme 3 the best current working model for describing the molecular mechanism of CFTR potentiation by Vx-770. Average rate constants obtained from ensemble fits to three pairs of current traces, such as those in *Figure 5f*, are summarized in *Table 2* (*left*).

Considering that measured diffusion coefficients of small molecules within a lipid bilayer fall in the range of $10^{-8}$ to $10^{-7}$ cm$^2$/s (*Ladha et al., 1996*; *Macháň and Hof, 2010*), the apparent second-order on-rates ($k_{on}$') of Vx-770 obtained from the fits to Scheme 3 are several orders of magnitude slower than expected for diffusion-limited binding of membrane-dissolved drug molecules to the CFTR protein. One possible explanation for this slow apparent on-rate could be that the drug remains associated with the polar surface of the bilayer, consistent with the hydrogen bonding capability of its five polar groups (*Figure 1a*). In that case, if the drug binding sites on the CFTR protein are located in the outer leaflet of the membrane, drug binding could be rate limited by slow flipping of the drug from the inner to the outer leaflet. Such a scenario could be consistent with the location of the two binding sites proposed by Yeh and colleagues, which are closer to the extracellular than to the intracellular membrane surface (*Yeh et al., 2019*). However, despite the introduction of an additional free parameter ($k_{flip}$), a two-compartment membrane model (Scheme 4, *Figure 5d*) only marginally improved the fits (*Figure 5f*, *red fit lines*). Thus, unless molecular dynamics simulations provide further support for such a model, we do not consider Scheme 4 (*Figure 5d*, *Table 2*, *right*) a substantially better working model compared to Scheme 3. An alternative possible explanation for the drug's slow apparent on-rate could be diffusion-limited loose binding followed by a slower induced fit conformational change.

Two common mechanistic conclusions suggested by the Scheme-3 and Scheme-4 fits are that (i) two drug molecules need to bind to the channel protein to achieve potentiation, and that (ii) the

**Table 2.** Model fit parameters for Schemes 3 and 4.

Average fit parameters (mean ± S.E.M.) obtained from global fits of Schemes 3 (*left column*) and 4 (*right column*) to three pairs of macroscopic WT CFTR current activation/deactivation time courses recorded in response to addition/removal of 0.01x saturated and 1x saturated concentrations of Vx-770 to the cytosolic (bath) solution. * Parameters $k_{in}$ and $k_{out}$ were constrained through $k_{in}/k_{out} = D_{oct/wat(pH=7.1)} = 62580$. ** Parameters $K_{d;mem}$, $K_{0.5;mem}$, and $K_{0.5;aq}$ were calculated from the fit parameters as described in Materials and methods.

| Scheme | 3 | 4 |
|---|---|---|
| $k_{in}$ (s$^{-1}$)* | 4034 ± 464 | 9794 ± 531 |
| $k_{out}$ (s$^{-1}$)* | 0.06446 ± 0.00742 | 0.1565 ± 0.0085 |
| $k_{on}$' (M$^{-1}$s$^{-1}$) | 870.3 ± 104.0 | 3703 ± 627 |
| $k_{off}$ (s$^{-1}$) | 0.007451 ± 0.000871 | 0.01075 ± 0.00009 |
| $k_{flip}$ (s$^{-1}$) | - | 0.04408 ± 0.00583 |
| $P_{o;max}/P_{o;bas}$ | 3.935 ± 0.001 | 3.996 ± 0.007 |
| | | |
| $K_{d;mem}$ (µM)** | 8.751 ± 1.160 | 3.106 ± 0.605 |
| $K_{0.5;mem}$ (µM)** | 21.13 ± 2.80 | 7.498 ± 1.460 |
| $K_{0.5;aq}$ (nM)** | 0.6752 ± 0.0895 | 0.6690 ± 0.0955 |

DOI: https://doi.org/10.7554/eLife.46450.009

slow channel deactivation time course following drug removal reflects the slow dissociation rate of the bound drug from the protein ($k_{off}$), rather than slow washout of unbound drug molecules from the membrane ($k_{out}$).

In conclusion, we have established the basic physicochemical properties of Vx-770, including its solubility profile and the thermodynamic parameters of the aqueous solution process. We have further determined its true potency towards WT, G551D, and ΔF508 CFTR, which is orders of magnitude higher than previously believed. Finally, we have identified a kinetic model which is suitable to describe its molecular mechanism. Our findings provide a solid framework to guide extensive current research efforts aimed at studying Vx-770 effects *in vitro*. A clinically relevant aspect of our findings is that half-maximal CFTR potentiation is achieved already by an ~100 fold dilution of a saturated Vx-770 solution. Thus, it might be worth testing whether drug doses lower than the currently approved dosage regimes are equally beneficial to CF patients, as this could reduce treatment costs that are currently ~300000 USD/patient/year.

## Materials and methods

### Key resources table

| Reagent type (species) or resource | Designation | Source or reference | Identifiers | Additional information |
|---|---|---|---|---|
| Biological sample (*Xenopus laevis*) | *Xenopus laevis* oocytes | African Reptile Park | RRID: NXR_0.0080 | mandyvorster@xsinet.co.za |
| Commercial assay or kit | HiSpeed Plasmid Midi Kit | Qiagen | 12643 | |
| Commercial assay or kit | QuickChange II Mutagenesis Kit | Agilent Technologies | 200524–5 | |
| Commercial assay or kit | mMESSAGE mMACHINE T7 Transcription Kit | ThermoFisher | AM1344 | |
| Chemical compound, drug | Collagenase type II | Gibco | 17107–0125 | |
| Chemical compound, drug | Adenosine 5'-triphosphoribose magnesium (ATP) | Sigma-Aldrich | A9187 | |
| Chemical compound, drug | Protein kinase A catalytic subunit, bovine | Sigma-Aldrich | P2645 | |
| Chemical compound, drug | Vx-770 (solid) | Selleck Chemicals | S1144 | |
| Chemical compound, drug | Vx-770 (10 mM in DMSO) | Selleck Chemicals | S1144 | |
| Software, algorithm | Pclamp9 | Molecular Devices | RRID: SCR_011323 | |

### Determination of Vx-770 solubility profile

Crystalline Vx-770 (purity 99.58% by HPLC) was purchased from Selleck Chemicals (Houston, TX, USA). Solubilities (S) at 25°C in anhydrous dimethyl sulfoxide (DMSO) (S ~ 0.77 M), 1-octanol (S ~ 3.9 mM), ethanol (S ~ 3.1 mM), octane (S ~ 2.6 mM), and pentane (S ~ 18 μM) were determined by weighing out a small amount (1–2 mg) of crystals on an analytical balance (Kern ABJ), and adding incremental small volumes of solvent, followed by mixing, until all crystals had been dissolved. However, even the smallest amount of Vx-770 that we could weigh out (~0.1 mg, obtained by manually dividing an ~0.5 mg aliqout into five comparable heaps under a stereomicroscope) failed to dissolve in 1 liter of water, as evident from clearly visible crystals even after vigorous shaking (200 RPM, New Brunswick Excella E24) for 24 hr at 25°C.

To obtain an aqueous solution (pH = 7.1) exactly saturated with Vx-770 (our '1x saturated' stock), a small aliquot of Vx-770 crystals (~0.1 mg) was added to 250 ml of our standard bath solution (in

mM: 134 NMDG-Cl, 2 $MgCl_2$, 5 HEPES, 0.5 EGTA, pH = 7.1 with NMDG), and the solution was shaken for 24 hr at 25°C at 200 RPM (New Brunswick Excella E24). Remaining, clearly visible, macro-crystals were removed by filtering the solution twice through an 0.22 μm pore size filter (Millipore Durapore Steritop + Stericup, Merck KGaA, Darmstadt, Germany). To sediment invisible microcrys-tals, the filtrate was centrifuged at 7200 RCF for 1 hr in 50 ml conical tubes, and the supernatant carefully removed, leaving the last ~1 ml behind. The resulting cleared filtrate was stored and used as a stock solution for all subsequent studies.

The concentration of Vx-770 in the 1x saturated stock was determined by spectrophotometry (NanoPhotometer P300, Implen GmbH). Vx-770 dissolved in 1-octanol shows strong absorption with a peak at 311 nm. To obtain a calibration curve, Vx-770 (from a commercial 10 mM stock dissolved in DMSO, purchased from Selleck Chemicals) was diluted into 1-octanol (saturated with water) to final concentrations of 1, 2, 4, 8, and 16 μM. Absorption spectra were measured in a 1 ml quartz cuvette against 1-octanol (saturated with water) as a blank (*Figure 1b*, *blue spectra*). Peak absorp-tion at 311 nm was linear with Vx-770 concentration across the entire range (the 0–4 μM range is shown in *Figure 1c*, *blue dots*). To extract Vx-770 from the 1x saturated aqueous stock, two 15 ml conical tubes, each containing 14 ml of the 1x aqueous stock plus 0.7 ml 1-octanol, were vortexed for 1 min, and then centrifuged at 7200 RCF for 30 min. Approximately 2 × 0.6 ml of the 1-octanol phase was recovered, pooled, and its absorption spectrum measured in a 1 ml quartz cuvette against 1-octanol (saturated with water) as a blank (*Figure 1b*, *red spectrum*). To extract the remain-ing aqueous phase for a second time, the remnants of the 1-octanol phase were carefully discarded from both conical tubes, and 12.5 ml of the remaining aqueous phase (removed by inserting the pipette tip well below the surface) was transferred into two clean conical tubes. The extraction was then repeated using 2 × 0.625 ml 1-octanol, from which 2x ~ 0.5 ml was recovered, pooled, and assayed for optical density (*Figure 1b*, *green spectrum*).

Aqueous solubility at pH = 7.4 (25°C) was measured as described above for pH = 7.1, but starting from a standard bath solution with pH adjusted to 7.4 using NMDG. The spectrophotometric calibra-tion curve was obtained in this case using 1, 2, 4, 8, and 16 μM Vx-770 dissolved in 1-octanol (satu-rated with the pH = 7.4 bath solution). The obtained calibration curve was identical to that shown in *Figure 1b–c* (i.e., Vx-770 fluorescence was insensitive to whether it was dissolved in 1-octanol satu-rated with water (as in *Figure 1*), or with saline buffered to either pH = 7.1, 7.3, or 7.4).

To determine aqueous solubility at 37°C (pH = 7.1), an aliquot of standard bath solution was heated to 37°C in a water bath. Because the $pK_a$ of HEPES is temperature dependent, heating to 37°C caused a decrease in pH by 0.17 units. The pH of the solution was therefore readjusted to 7.1 at 37°C using NMDG. To obtain a 1x saturated solution at 37°C, overnight shaking with Vx-770 crystals, subsequent 2x filtering, and centrifugation were all perfomed at 37°C. To measure Vx-770 concentra-tion in the 1x saturated stock solution, 0.7 ml 1-octanol was added to 14 ml aliquots of the stock at 37°C, the tubes were vortexed for 1 min, and the subsequent steps of the 1-octanol extraction pro-cedure, as well as the spectrophotometric determination, were performed at room temperature, as described above.

## Estimation of thermodynamic parameters of the Vx-770 aqueous solution process

The standard enthalpy of solution was calculated as $\Delta H^{\circ}_{sol} = ((RT_1T_2)/(T_2-T_1))\cdot\ln(S(T_2)/S(T_1))$, where R = 8.31 $Jmol^{-1}K^{-1}$, $T_1$ = 298 K (25°C), $T_2$ = 310 K (37°C), and $S(T_1)$ and $S(T_2)$ are the measured aque-ous solubilities of Vx-770 at the respective temperatures (pH = 7.1). The standard free energy of solution at T = 298 K (25°C) was calculated as $\Delta G^{\circ}_{sol} = -RT\cdot\ln(S(T))$.

## Molecular biology

The G551D and ΔF508 mutations were introduced into pGEMHE-CFTR using the QuikChange Kit (Agilent, Santa Clara, CA, USA), and confirmed by automated sequencing; cDNA was purified (HiSpeed Plasmid Midi Kit, Qiagen), transcribed *in vitro* (mMessage T7 Kit, ThermoFisher Scientific, Waltham, MA USA) and cRNA stored at −80°C.

## Isolation and injection of *Xenopus laevis* oocytes

Oocytes were extracted from anaesthethized adult female *Xenopus laevis* (RRID: NXR_0.0080) following Institutional Animal Care Committee guidelines, isolated using collagenase treatment (Gibco, Collagenase type II), injected with 0.1–10 ng cRNA in a fixed 50 nl volume, and stored at 18°C in a modified frog Ringer's solution (in mM: 82 NaCl, 2 KCl, 1 $MgCl_2$, and 5 HEPES, pH 7.5 with NaOH) supplemented with 1.8 mM $CaCl_2$ and 50 µg/ml gentamycin. Current recordings were obtained 1–3 days after injection.

## Excised inside-out patch recording

Patch pipette solution contained (in mM): 136 NMDG-Cl, 2 $MgCl_2$, 5 HEPES, pH = 7.4 with NMDG. Bath solution contained (in mM): 134 NMDG-Cl, 2 $MgCl_2$, 5 HEPES, 0.5 EGTA, pH = 7.1 with NMDG. MgATP (2 mM) was added from a 400 mM aqueous stock solution (pH = 7.1 with NMDG). Before each experiment 300 nM catalytic subunit of PKA (Sigma-Aldrich Kft., Budapest, Hungary) was applied for ~1 min to phosphorylate CFTR channels. Vx-770 (Selleck Chemicals) was diluted into the bath solution either from a '1x saturated' aqueous stock solution (see above, concentration ~62 nM), or from a commercially obtained 10 mM stock dissolved in DMSO (Selleck Chemicals). Following patch excision the patch pipette was moved into a flow chamber, and recordings were done under continuous superfusion of the cytosolic patch surface. Solution compositions were exchanged using computer-driven electronic valves (solution exchange time constant <50 ms). After each recording day perfusion tubings were extensively washed with distilled water. Although supersaturated solutions of Vx-770 were not employed here, any perfusion tubing that had ever been in contact with a given concentration of Vx-770 was subsequently used only for solutions that contained the same, or higher, concentrations of the drug. After every 5–6 experimental days the entire perfusion system was replaced. Currents were recorded at 25°C, at a membrane potential of −40 mV, digitized at 10 kHz, Gaussian-filtered at 2 kHz, and recorded to disk (Axopatch 200B, Digidata 1322A, Pclamp9 (Molecular Devices, RRID: SCR_011323)). For display purposes, currents are shown filtered at 10 Hz and sampled at 20 Hz, and expanded single-channel traces (*Figure 2b*, *Figure 4d*) are shown filtered at 50 Hz and sampled at 250 Hz.

## Analysis of macroscopic current recordings

Vx-770 was applied after currents elicited by 2 mM ATP had stabilized. To allow for full development of its potentiator effect, Vx-770 was superfused until the current reached steady state which, for low nanomolar Vx-770, required up to 5–6 min. After full stimulation, Vx-770 was washed off for at least 5 min to allow the current to decline to a post-drug steady state. Fractional current stimulation by Vx-770 ($I/I_{control}$) was calculated as the mean steady current in the presence of the drug, divided by the average of the pre- and post-drug mean steady currents. Dose response curves (*Figure 3c,f*) were least-squares fitted to the modified Hill equation $I/I_{control}=(K_{0.5}^n+(I_{max}/I_{control})[Vx\text{-}770]^n)/(K_{0.5}^n+[Vx\text{-}770]^n)$, with $I_{max}/I_{control}$, $K_{0.5}$, and $n$ as free parameters.

To obtain apparent deactivation rates, current decay time courses following Vx-770 removal were fitted by single exponentials using least squares, and deactivation rates defined as the inverse of the decay time constant ($1/\tau_{off}$; *Figure 4c*). To obtain apparent activation rates, current activation time courses upon Vx-770 addition were fitted to an empirical three-step kinetic scheme $S_1{\rightarrow}S_2{\rightarrow}S_3{\rightarrow}S_4$. Compound states $S_1$-$S_4$ in this scheme are not interpreted as channel conformational states, but rather as states of the entire system (aqueous solution +patch membrane leaflets + channels), and could reflect various stages of drug distribution among those compartments. The open probability ($P_o$) of CFTR channels in compound states $S_1$, $S_2$, and $S_3$ was fixed to the $P_o$ observed for single WT CFTR channels under control conditions (~0.16; *Figure 2c*), whereas the $P_o$ for CFTR channels in compound state $S_4$, and transition rates $k_{12}$, $k_{23}$, and $k_{34}$, were left free (*Figure 4a*). (Whereas this simple irreversible scheme is clearly unsuitable to explain reversibility of drug effects, and does not address the mechanism of the drug, it provided sufficient flexibility for a perfect fit of individual activation time courses.) Apparent activation time constants ($\tau_{on}^*$) were defined as $T_{1/2}/\ln2$, where $T_{1/2}$ is the time required for the current to cross the midpoint between its pre-activated and fully activated amplitudes. Apparent activation rates were defined as $1/\tau_{on}^*$ (*Figure 4b*).

## Kinetic analysis of microscopic patches

Segments of current recording originating from 1 to 7 active channels were digitally filtered at 50 Hz, and idealized by half-amplitude threshold crossing. Steady-state open probabilities in 2 mM ATP, before or during application of Vx-770 or after its removal (*Figure 2c*), were calculated from the events lists as the time-average of the fraction of open channels. A closed-open-blocked (C-O-B) kinetic scheme, which separates brief (~10 ms) flickery closures (to state B) from long (~1 s) interburst closures (to state C), was fitted by maximum likelihood to the set of dwell-time histograms for all conductance levels, to obtain microscopic transition rates $r_{CO}$, $r_{OC}$, $r_{OB}$, and $r_{BO}$, while accounting for a fixed dead time of 6 ms (*Csanády, 2000*). The software is freely available upon request. Mean burst ($\tau_b$) and interburst ($\tau_{ib}$) durations were calculated as $\tau_b = (1/r_{OC})(1 + r_{OB}/r_{BO})$ and $\tau_{ib} = 1/r_{CO}$, and channel opening and closing rates (*Figure 2c*) defined as $1/\tau_{ib}$ and $1/\tau_b$, respectively. The stability plot in *Figure 4d* was obtained by calculating open probability over sequential 46 s time windows.

## Ensemble fitting of gating models to pairs of on-off current time courses

The dynamic equations that describe the kinetic models depicted in *Figure 5* are as follows:

### Scheme 1

$$dx_1(t)/dt = k_{in}\cdot V_c - 2\cdot k_{out}\cdot x_1(t) - k_{on}'\cdot x_1(t)\cdot x_2(t) + k_{off}\cdot(C_t - x_2(t))$$
$$dx_2(t)/dt = - k_{on}'\cdot x_1(t)\cdot x_2(t) + k_{off}\cdot(C_t - x_2(t))$$

where $x_1(t)$ is the concentration of free Vx-770 in the membrane, $x_2(t)$ is the concentration of unliganded drug binding sites (CFTR channels) in the membrane, $C_t$ is the total concentration of drug binding sites (CFTR channels) in the membrane, and $V_c$ is the concentration of Vx-770 in the cytosol.

### Scheme 2

$$dx_1(t)/dt = k_{in}\cdot V_c - (k_{out} + k_{flip})\cdot x_1(t) + k_{flip}\cdot x_2(t)$$
$$dx_2(t)/dt = k_{flip}\cdot x_1(t) - (k_{out} + k_{flip})\cdot x_2(t) - k_{on}'\cdot x_2(t)\cdot x_3(t) + k_{off}\cdot(C_t - x_3(t))$$
$$dx_3(t)/dt = - k_{on}'\cdot x_2(t)\cdot x_3(t) + k_{off}\cdot(C_t - x_3(t))$$

where $x_1(t)$ is the concentration of Vx-770 in the inner membrane leaflet, $x_2(t)$ is the concentration of free Vx-770 in the outer membrane leaflet, $x_3(t)$ is the concentration of unliganded drug binding sites (CFTR channels) in the outer membrane leaflet, $C_t$ is the total concentration of drug binding sites (CFTR channels) in the outer membrane leaflet, and $V_c$ is the concentration of Vx-770 in the cytosol.

### Scheme 3

$$dx_1(t)/dt = k_{in}\cdot V_c - 2\cdot k_{out}\cdot x_1(t) - k_{on}'\cdot(2\cdot x_2(t) + x_3(t))\cdot x_1(t) + k_{off}\cdot(2\cdot(C_t - x_2(t) - x_3(t)) + x_3(t))$$
$$dx_2(t)/dt = - 2\cdot k_{on}'\cdot x_1(t)\cdot x_2(t) + k_{off}\cdot x_3(t)$$
$$dx_3(t)/dt = 2\cdot k_{on}'\cdot x_1(t)\cdot x_2(t) + 2\cdot k_{off}\cdot(C_t - x_2(t) - x_3(t)) - (k_{off} + k_{on}'\cdot x_1(t))\cdot x_3(t)$$

where $x_1(t)$ is the concentration of free Vx-770 in the membrane, $x_2(t)$ is the concentration of unliganded CFTR channels in the membrane, $x_3(t)$ is the concentration of monoliganded CFTR channels in the membrane, $C_t$ is the total concentration of CFTR channels in the membrane, and $V_c$ is the concentration of Vx-770 in the cytosol.

### Scheme 4

$$dx_1(t)/dt = k_{in}\cdot V_c - (k_{out} + k_{flip})\cdot x_1(t) + k_{flip}\cdot x_2(t)$$
$$dx_2(t)/dt = k_{flip}\cdot x_1(t) - (k_{out} + k_{flip})\cdot x_2(t) - k_{on}'\cdot(2\cdot x_3(t) + x_4(t))\cdot x_2(t) + k_{off}\cdot(2\cdot(C_t - x_3(t) - x_4(t)) + x_4(t))$$
$$dx_3(t)/dt = - 2\cdot k_{on}'\cdot x_2(t)\cdot x_3(t) + k_{off}\cdot x_4(t)$$
$$dx_4(t)/dt = 2\cdot k_{on}'\cdot x_2(t)\cdot x_3(t) + 2\cdot k_{off}\cdot(C_t - x_3(t) - x_4(t)) - (k_{off} + k_{on}'\cdot x_2(t))\cdot x_4(t)$$

where $x_1(t)$ is the concentration of Vx-770 in the inner membrane leaflet, $x_2(t)$ is the concentration of free Vx-770 in the outer membrane leaflet, $x_3(t)$ is the concentration of unliganded CFTR channels in the outer membrane leaflet, $x_4(t)$ is the concentration of monoliganded CFTR channels in the outer

membrane leaflet, $C_t$ is the total concentration of CFTR channels in the outer membrane leaflet, and $V_c$ is the concentration of Vx-770 in the cytosol.

To calculate the value of $C_t$, the total number of channels in the patch (N) was estimated by assuming $P_{o;bas}=0.16$ for prephosphorylated channels in ATP (cf., **Figure 2c**). For Schemes 1 and 3 the volume of the membrane patch was taken as $10^{-16}$ liter (assuming a macroscopic patch surface area of ~4 $\mu m^2$ increased by ~5 fold at a microscopic level due to the presence of microvilli (**Dascal, 1987**), and a membrane thickness of ~5 nm), and for Schemes 2 and 4 the volume of the outer membrane leaflet was taken as $5 \cdot 10^{-17}$ liter. $C_t$ was then calculated as $N/N_A$ divided by the relevant volume ($N_A$, Avogadro's number). (In control trials in which up to 5-fold larger/smaller fixed membrane volumes were used, the fits showed little sensitivity to the choice of membrane volume, apart from appropriate rescaling of the obtained values for $k_{on}'$ and $k_{flip}$.)

For the ensemble fitting, all experimental current traces were normalized to their steady-state values observed prior to exposure to Vx-770. Predicted normalized on- and off- current time courses for each model were calculated from the above dynamic equations by solving for the time-dependent evolution of the vector **x**, using a modified Euler's method (**Press, 1992**). For the on- time courses the initial values for all $x_i$ were set to zero, for the off- time courses the final **x** vector of the preceding on- time course served as the initial vector while $V_c$ was set to zero. The predicted time course of the normalized current trace was then obtained as $x_2(t)/C_t + (1-x_2(t)/C_t)\cdot(P_{o;max}/P_{o;bas})$ (Scheme 1), $x_3(t)/C_t + (1-x_3(t)/C_t)\cdot(P_{o;max}/P_{o;bas})$ (Scheme 2), $(x_2(t)+x_3(t))/C_t + (1-(x_2(t)+x_3(t))/C_t)\cdot(P_{o;max}/P_{o;bas})$ (Scheme 3), and $(x_3(t)+x_4(t))/C_t + (1-(x_3(t)+x_4(t))/C_t)\cdot(P_{o;max}/P_{o;bas})$ (Scheme 4), respectively. The sets of predicted normalized on-off time courses were fitted to the sets of normalized experimental current traces by simple least squares, using a downhill simplex method for optimization (**Press, 1992**).

## Steady-state solutions of the models and calculation of predicted apparent affinities

Under the conditions used here, that is upon prolonged exposure of the cytosolic face of the patch to a constant concentration of Vx-770, with no Vx-770 present on the extracellular side, all four kinetic models shown in **Figure 5** predict relaxation of the system to a steady state. At steady state there is a constant outward directed transmembrane flux of Vx-770, while its concentration remains constant within the membrane (or membrane leaflets), and close to zero in the extracellular (pipette) solution. (The volume of the pipette solution, on the order of $10^{-5}$ l, can be considered infinitely large relative to the volume of the patch membrane). For Schemes 1 and 3, the steady-state membrane concentration of the drug is $V_{m;\infty}=V_c\cdot k_{in}/(2\cdot k_{out})$. For Schemes 2 and 4, the steady-state drug concentrations in the cytosolic and external membrane leaflets ($V_{mc;\infty}$, $V_{me;\infty}$), respectively, are given by $V_{mc;\infty}=V_c\cdot(k_{in}/k_{out})\cdot(k_{out}+k_{flip})/(k_{out}+2\cdot k_{flip})$ and $V_{me;\infty}=V_c\cdot(k_{in}/k_{out})\cdot k_{flip}/(k_{out}+2\cdot k_{flip})$.

Once the drug concentrations in the various compartments have stabilized, the channel pool approaches an equilibrium distribution between its available (unliganded, ligandbound, and, for Schemes 3 and 4, diliganded) forms. The concentration-dependence of steady-state current activation reflects the concentration-dependence of the fractional occupancy of the monoliganded state ($Y_1$) for Schemes 1 and 2, but of the diliganded state ($Y_2$) for Schemes 3 and 4. The latter are given by $Y_1 = V_\infty/(K_{d;mem}+V_\infty)$ and $Y_2 = (V_\infty/(K_{d;mem}+V_\infty))^2$, respectively, where $K_{d;mem} = k_{off}/k_{on}'$, and $V_\infty=V_{m;\infty}$ for Schemes 1 and 3, but $V_\infty=V_{me;\infty}$ for Schemes 2 and 4. The midpoints of these predicted concentration response curves ($K_{0.5;mem}$), that is the drug concentrations in the target compartment (membrane or external membrane leaflet) that cause half-maximal stimulation, are obtained as $K_{0.5;mem} = K_{d;mem}$ for Schemes 1 and 2, but $K_{0.5;mem} = (1+\sqrt{2})\cdot K_{d;mem}$ for Schemes 3 and 4. When plotted using a logarithmic abscissa, the midpoint slope of the dose response curve for Schemes 3 and 4 corresponds to that of a Hill function with Hill coefficint $n_H = 2\cdot(2-\sqrt{2})$. The apparent drug affinities ($K_{0.5;aq}$), that is the drug concentrations in the cytosolic solution that cause half-maximal stimulation, are obtained as $K_{0.5;aq} = K_{0.5;mem}/(k_{in}/(2\cdot k_{out}))$ for Schemes 1 and 3, but $K_{0.5;aq} = K_{0.5;mem}/((k_{in}/k_{out})\cdot k_{flip}/(k_{out}+2\cdot k_{flip}))$ for Schemes 2 and 4.

## Statistics

Data are presented as mean ± S.E.M from 3 to 13 independent measurements, as indicated in the figure legends. Statistical significance was quantified using Student's two-tailed $t$ test, differences are reported as not significant for $p > 0.05$, and significant for $p < 0.05^*$ or $p < 0.01^{**}$.

## Acknowledgements

Supported by MTA Lendület grant LP2017-14/2017 and Cystic Fibrosis Foundation Research Grant CSANAD17G0.

## Additional information

### Competing interests

László Csanády: Reviewing editor, *eLife*. The other author declares that no competing interests exist.

### Funding

| Funder | Grant reference number | Author |
| --- | --- | --- |
| Cystic Fibrosis Foundation | CSANAD17G0 | László Csanády |
| Magyar Tudományos Akadémia | Lendület grant LP2017-14/2017 | László Csanády |

The funders had no role in study design, data collection and interpretation, or the decision to submit the work for publication.

### Author contributions

László Csanády, Conceptualization, Formal analysis, Funding acquisition, Investigation, Methodology, Writing—original draft, Project administration, Writing—review and editing; Beáta Töröcsik, Generated mutant CFTR constructs, Performed in vitro transcription, Purified cRNA

### Author ORCIDs

László Csanády https://orcid.org/0000-0002-6547-5889

### Ethics

Animal experimentation: This study was performed in strict accordance with the recommendations in the Guide for the Care and Use of Laboratory Animals of the National Institutes of Health. All of the animals were handled according to approved institutional animal care and use committee (IACUC) protocols of Semmelweis University (last approved 06-30-2016, expiration 06-30-2021).

### Decision letter and Author response

Decision letter https://doi.org/10.7554/eLife.46450.012
Author response https://doi.org/10.7554/eLife.46450.013

## Additional files

### Supplementary files

• Transparent reporting form
DOI: https://doi.org/10.7554/eLife.46450.010

### Data availability

All data generated or analyzed during this study are included in the manuscript or can be visualized in the figures.

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
