## [Decision Letter]

[Editors’ note: this article was originally rejected after discussions between the reviewers, but the authors were invited to resubmit after an appeal against the decision.]

Thank you for submitting your work entitled "Cystic fibrosis drug ivacaftor stimulates CFTR channels at picomolar concentrations" for consideration by *eLife*. Your article has been reviewed by three peer reviewers, including Leon D Islas as the Reviewing Editor and Reviewer #1, and the evaluation has been overseen by a Senior Editor. The following individual involved in review of your submission has agreed to reveal their identity: Sidney Simon (Reviewer #3).

Our decision has been reached after consultation between the reviewers. Based on these discussions and the individual reviews below, we regret to inform you that your work will not be considered further for publication in *eLife*.

While the reviewers agree that the data presented in the manuscript is sound and represents an important technical development in understanding the action of ivacaftor on CFTR, they also think that the paper did not provide any insight into the molecular interaction of the drug with the channel and the determinants of solubility in biological tissues. The reviewers point out that this diminishes the relevance and interest of the paper to a broader readership.

*Reviewer #1:*

The CFTR Cl^-^ ion channel is responsible for adequate mucus secretion in respiratory epithelia. Multiple mutations are associated with CFTR dysfunction and consequent induction of cystic fibrosis (CF), which can be lethal. One of the few treatments approved for CF is the drug Vx-770, which has been shown to be effective in potentiating CFTR mutants characterized by a low open probability, a gating defect.

In this paper by Csanády and Töröcsik present data that resolve a standing controversy in the CFTR and Vx-770 literature and which could also have an impact in the clinical application of this drug to CF patients.

The experiments included in this manuscript show that Vx-770 has an extremely low solubility in water. As a consequence of this, the dose utilized in diverse laboratories is likely extremely high and mostly undetermined, accounting for the different values of EC_50_ reported. More interestingly, Vx-770 effects are reportedly irreversible. This manuscript presents a determination of the solubility of Vx-770 in several solvents, including water, and shows that when the correct activating doses (below nM concentrations) are applied, the drug effects are perfectly reversible.

The experiments are of high quality and represent a solid foundation to start to understand the biophysics and molecular basis of the gating effects of this drug.

In addition, this is an important contribution that might have clinical relevance, as it will help understand the pharmacokinetics of Vx-770.

I have a suggestion regarding the use of the multiple-step model employed to explain the delay in activation time course when Vx-770 is applied. The suggested model is an irreversible model, which may appear as a contradiction with the finding that the drug effects are indeed reversible. Also, it should be clear what state occupancy is being compared with the current time course. And finally it should be clearly stated that this is not a scheme that explain gating, that is, states S1 to S4 are not to be interpreted as channel conformational states.

*Reviewer #2:*

In this study the authors determine the solubility profile and stimulatory potency of Vx-770 (ivacaftor), a CFTR potentiator drug in clinical use to treat people with cystic fibrosis, towards wild-type and mutant human CFTR channels using the patch-clamp technique with excised inside-out membrane patches. They find that its solubility in aqueous saline (pH=7.1) is ~200-fold lower (~60 nanomolar), whereas the potency of its stimulatory effect is >100-fold higher, than previously reported, and is fully reversible. They find evidence that several slow steps are involved in the activation pathway. The experiments are generally well done and of high quality. The manuscript is well written. However, significance is reduced because no new insight is gained how Vx-770 interacts with CFTR to activate channel activity. Usually a methodological progress is presented in the context of describing an important new insight that could be obtained because of the method. In addition, because solubility in biological fluids at physiological pH and body temperature is not studied, the results appear only relevant to a small group of readers who perform CFTR patch clamp electrophysiology with excised membrane patches.

In addition, I have the following comments:

In subsection “CFTR channel stimulation by Vx-770 is fully reversible”: the authors state "up to ~200-fold higher than those that can be ever attained in the human body using oral drug administration." This statement appears not justified by the data. The authors do not study solubility of Vx-770 in biological fluids and tissues. For instance, protein present in a biological intra- or extracellular fluid may increase solubility substantially. In this context it is also not clear to me why the authors do not test solubility at 37°C and pH values that resemble intra- or extracellular pH.

Figure 2C: A one-tailed Student t-test appears not appropriate. First, since CFTR channel inactivation by Vx-770 has been described among the effects of Vx-770 in the literature (see Introduction section final paragraph) the statistical test should be two-tailed. Second, repeated measurement ANOVA should be considered since technically there are two interventions with the same channel.

*Reviewer #3:*

In this paper the WT and two mutants of the CTFR channel are investigated in inside out patches in the absence and presence of continually perfused Vx-770, a clinically relevant drug used in CF patients. In my opinion the most important finding of this work is that they found is that, in contrast to previous studies that found Vx-770 to be irreversible, they found that by using lower concentrations that it is completely reversible. They measured its oil (octanol etc.)/water (saline) partition coefficient although I think it would have been more useful to measure the membrane (GUV's) +/- cholesterol/saline partition coefficient given they were testing the drug in frog oocytes. Moreover, since they wrote "the basic properties of the drug remain unknown or controversial", I think a description of the drug, its shape (planar), purity (important as K1/2 = 0.47 nM) and its pK_a_ values in salt and low dielectric media (if known, and if not discuss) would be useful to the reader. Also, I would like to know (if known) where on the membrane or channel is it binding and if its action a consequence of membrane deformation and/or to a binding site on CTFR that has not been alluded to. In their methods they vortexed the Xstals but I wonder if they ever sonicated and/or heated them to get them into solution.

[Editors’ note: what now follows is the decision letter after the authors’ article was reconsidered after an appeal.]

Thank you for submitting your article "Cystic fibrosis drug ivacaftor stimulates CFTR channels at picomolar concentrations" for consideration by *eLife*. Your article has been reviewed by Leon Islas as Reviewing Editor and Reviewer #1, and the evaluation has been overseen Richard Aldrich as the Senior Editor.

The editors have discussed the appeal with one another and are happy to consider a revised version of your manuscript.

Essential revisions:

The paper by Csanády and Töröcsik describes in detail the solubility of the clinically important drug ivacaftor, an activator of CFTR channels. This is an appeal of a previous decision and the author's proposed changes for improvement are reasonable and will improve the paper. Our suggestion is that you make the changes that are proposed in your appeal letter, especially place more emphasis on the clinical *and* physiological relevance of the findings. Pay attention to the suggestion of measuring solubility at 37°C and make a point of clarifying the applicability of the results to the context of biological fluids. A discussion of the implication of the results for the pharmacokinetics of ivacaftor can also help to further the case for the clinical relevance of the present findings.

---

## [Author Response]

[Editors’ note: the author responses to the first round of peer review follow.]

I kindly ask for the reevaluation of the reasons that have led to the rejection of our manuscript, and for a potential reconsideration of the editorial decision. The Reviewers all agreed that the results support the conclusions. The rejection was based on two issues: (1) a lack of insight into the molecular interaction of the drug with the channel, and (2) a lack of information on its solubility in biological tissues. I have serious objections against both of these major criticisms, as discussed in detail below.

1) "significance is reduced because no new insight is gained how Vx-770 interacts with CFTR to activate channel activity. Usually a methodological progress is presented in the context of describing an important new insight that could be obtained because of the method." (Reviewer 2)

As to the question of significance, I had sent a presubmission inquiry to Senior Editor Richard Aldrich, describing him the main findings of this study, and explicitly pointing out that no biological mechanisms are addressed. In his response, Dr. Aldrich judged the findings of sufficient interest, and encouraged submission to *eLife*. It comes therefore as a surprise that the findings are judged insignificant at the present stage. I do agree that in general determination of solubility and potency of a drug is in itself not a major advance. But there are not many drugs that make it to the clinics, and ivacaftor is one of them. The fact that ~100000 CF patients are taking the drug on a daily basis warrants that studying its mechanisms will remain in the center of research efforts for years to come. On the other hand, despite a very large number (>500) of studies published on the drug since its discovery, both its binding site on the protein and its mechanism remain elusive, in large part due to inadequately designed experiments in which highly supersaturated solutions were used. Therefore, I do think that in the present case the simple establishment of the drug's true solubility profile, true potency, and concentration dependence of its apparent on- and off-rates, is a major breakthrough which will put all published studies into a new perspective, and allow all future studies to be planned adequately.

As to the second point, reviewers 2 and 3 seem to have misunderstood the relevance of a saturated vs. supersaturated solution, and they have confounded the relevance of drug-protein binding with that of true solubility. In truth − except for a potential effect on solubility of elevating temperature to 37^o^C (which we could verify) − all our results apply to the biological fluids and cell membranes of a living organism, as discussed below.

2.1) "solubility in biological fluids at physiological pH and body temperature is not studied" and"the authors state "up to ~200-fold higher than those that can be ever attained in the human body using oral drug administration." This statement appears not justified by the data. The authors do not study solubility of Vx-770 in biological fluids and tissues. For instance, protein present in a biological intra- or extracellular fluid may increase solubility substantially." (Reviewer 2)

Of course, it is well known that Vx-770 binds to plasma proteins, and total plasma concentrations in patients following drug administration have been measured and documented by the FDA: peak total plasma concentrations may reach values up to 10 μM. However, cell membranes are at equilibrium with FREE dissolved Vx-770, not with the protein-bound drug. Therefore, in a living organism the target cell membranes will accumulate Vx-770 only to an extent that is at equilibrium with FREE dissolved Vx-770 (in the extracellular fluid and/or in the cytosol). Thus, although drug binding proteins are important for pharmacokinetics (they increase the drug transport capacity of the blood and therefore oral bioavailability, and act as a drug buffer system that replenishes the free drug pool during the time course of its elimination from the body), for the effective drug concentration in the target cell membrane the only relevant parameter is FREE dissolved Vx-770 − for which we have established an upper limit here.

2.2) "In this context it is also not clear to me why the authors do not test solubility at 37°C and pH values that resemble intra- or extracellular pH." (Reviewer 2)

The composition of the aqueous saline in which Vx-770 solubility was determined resembled that of an intracellular fluid with respect to ionic strength, osmolarity, as well as pH (7.1). We did not test solubility at 37^o^C, but this could be very easily added. Of note, although solubility at 37^o^C is clearly of academic interest, we believe that solubility at 25^o^C is practically more relevant: solubility has to be known under the conditions that are used in experiments aimed at understanding the mechanisms of the drug. The majority of such experiments have been conducted at room temperature.

2.3) "I am sure Vx-770 will bind to various biological structures and proteins when it is in the body so I doubt it at supersaturated concentrations in the fluid phase." (Reviewer 3)

This is a misunderstanding. We did not claim that supersaturated concentrations are reached in the body. Quite on the contrary, we claim that in the body FREE drug concentrations will never become higher than saturated.

2.4) "Is there any evidence that Vx-770 will form crystalline precipitates in the bilayer?" (Reviewer 3)

This is a thermodynamic necessity. When a saturated aqueous solution is at equilibrium with a membrane, the membrane also has to be saturated. Similarly, when a membrane is exposed to a supersaturated solution (that is, a solution which is kept supersaturated by continuous superfusion), the membrane will also become supersaturated. Thus, over time crystals will have to form. Indirect experimental evidence comes from our demonstration that Vx-770 effects are fully reversible when supersaturating concentrations are avoided, whereas they are reportedly irreversible when supersaturated solutions are used.

2.5) "by definition a solution is saturated when it is in equilibrium with its bulk phase (i.e. Xstals)- it is not necessarily supersaturated then." (Reviewer 3)

We agree. This is exactly how we generated our 1x saturated stock: by equilibrating the solution with crystals, followed by removal of undissolved crystals. In contrast, in previous studies supersaturated solutions were obtained by diluting Vx-770 from a DMSO-based stock into an aqueous buffer. Because Vx-770 is 10-million times more soluble in DMSO compared to water, it is easy to generate supersaturated aqueous solutions using the latter approach.

3) The remaining comments are all issues that could be easily amended as follows:

3.1) "I have a suggestion regarding the use of the multiple-step model employed to explain the delay in activation time course when Vx-770 is applied. The suggested model is an irreversible model, which may appear as a contradiction with the finding that the drug effects are indeed reversible. Also, it should be clear what state occupancy is being compared with the current time course. And finally it should be clearly stated that this is not a scheme that explain gating, that is, states S1 to S4 are not to be interpreted as channel conformational states." (Reviewer 1)

The reviewer is absolutely right that the modeled states are not interpreted as channel conformational states, but rather as states of the entire system (aqueous solution + patch membrane leaflets + channels), and likely reflect various stages of drug distribution between those compartments. It is also true that in its present (irreversible) form the fitted scheme seems to contradict the demonstrated reversibility of the drug effects, and so allowing reversibility for all three steps would seem logical. However, in the absence of more specific structural information, we feel that postulation of any specific mechanistic model would be highly speculative. We therefore simply aimed to find the mathematical function with the smallest number of free parameters suitable to describe the on- time courses, and used the fitted curves only to obtain the apparent on- time constant (τ*). All this could be of course stated and explained much more clearly in Materials and methods.

3.2) "Figure 2C: A one-tailed Student t-test appears not appropriate. First, since CFTR channel inactivation by Vx-770 has been described among the effects of Vx-770 in the literature (see Introduction section final paragraph) the statistical test should be two-tailed. Second, repeated measurement ANOVA should be considered since technically there are two interventions with the same channel." (Reviewer 2)

This can be easily amended.

3.3) "They measured its oil (octanol etc.)/water (saline) partition coefficient although I think it would have been more useful to measure the membrane (GUV's) +/- cholesterol/saline partition coefficient given they were testing the drug in frog oocytes." (Reviewer 3)

The primary purpose of this study was to determine the aqueous solubility of the drug. Solubilities in ethanol or DMSO were added for practical reasons, since those solvents are used by researchers to generate stock solutions of the drug. Solubility in octanol, and the octanol:water partition coefficient, were determined for three reasons. First, the octanol:water distribution coefficient is a very good predictor of a drug's overall membrane solubility, and is one of the key parameters used in the pharmaceutical industry to compare drugs. Second, octanol is immiscible with water, and can therefore be conveniently used to extract the drug from the aqueous phase. Third, Vx-770 dissolved in octanol is a well defined, homogeneous, single-phase system (a true solution) in which the drug concentration can be readily measured using spectrophotometry. The latter two properties were exploited to determine the aqueous solubility of the drug. In contrast, GUV's (i) represent a poorly defined system (with properties highly dependent on lipid composition), (ii) form a colloidal suspension in water rather than a homogeneous system, and (iii) determination of drug solubility in GUV's would be technically challenging.

3.4) "a description of the drug, its shape (planar), purity (important as K1/2 = 0.47 nM) and its pK_a_ values in salt and low dielectric media (if known, and if not discuss) would be useful to the reader." (Reviewer 3)

Formula and shape can be added to Figure 1. Purity (99.58% by HPLC by manufacturer's certificate of analysis) can be added to Materials and methods. Obtaining pK_a_ values experimentally would be challenging given low nanomolar aqueous solubility. Computational predictions yield pK_a_ values that scatter broadly, depending on the algorithm used (e.g., Chemaxon Chemicalize, Marvin). This information can be added.

3.5) "Also, I would like to know (if known) where on the membrane or channel is it binding and if its action a consequence of membrane deformation and/or to a binding site on CTFR that has not been alluded to." (Reviewer 3)

Neither the binding site of the drug on the channel protein, nor its mechanism of action is currently known.

3.6) "In their methods they vortexed the Xstals but I wonder if they ever sonicated and/or heated them to get them into solution." (Reviewer 3)

Sonication is primarily useful for the dispersion of particles in a solvent to generate a colloid (dispersion, suspension or emulsion), such as a preparation of micelles from amphipathic detergent molecules, or unilamellar vesicles from membrane lipids. It is true that sonication can also speed up dissolution of small molecules, however, it can easily result in the generation of supersaturated solutions. Therefore, for the determination of true solubility sonication is better avoided. Our approach of vigourous shaking of drug crystals in small volumes (250 ml) of buffer for 24 hours seems a safer approach for reaching thermodynamic equilibrium. Of note, in four independent experiments we obtained identical final aqueous drug concentrations (62 ± 2 nM) following removal of undissolved crystals (by filtering + centrifugation), although the total amount of added drug was not identical, as the crystals were not weighed out on a balance. This indicates that in all four cases saturation was achieved.

Heating is clearly irrelevant for the determination of drug solubility at 25^o^C. For academic reasons, solubility at 37^o^C could be measured as an additional piece of information. See also response to comment 2.2 above.

[Editors’ note: the author responses to the re-review follow.]

Essential revisions:The paper by Csanády and Töröcsik describes in detail the solubility of the clinically important drug ivacaftor, an activator of CFTR channels. This is an appeal of a previous decision and the author's proposed changes for improvement are reasonable and will improve the paper.

We thank the editors for giving us a chance to address the concerns raised in the original reviews. We have extensively revised the manuscript along three major lines:

i) We have measured pH-dependence (in the physiological pH range), as well as temperature dependence of aqueous solubility of Vx-770, and calculated thermodynamic parameters of the solution process. We find that solubility is not increased when pH is elevated from 7.1 (intracellular pH) to 7.4 (extracellular pH). This implies that the smallest acidic pK_a_ value of the drug is substantially higher than 7.4. Therefore, in physiological fluids, the major microspecies of the drug is the fully protonated uncharged form. Thus, the distribution of the drug between the extra- and intracellular space is not influenced by the transmembrane pH gradient or the membrane potential (subsection “pH- and temperature dependence of Vx-770 solubility” first paragaph). We further find an ~2-fold higher aqueous solubility of the drug at 37^o^C relative to 25^o^C, which is consistent with reported temperature dependences of solubilities of other highly hydrophobic drugs, and offers a thermodynamic explanation for the low solubility of Vx-770 (subsection “pH- and temperature dependence of Vx-770 solubility” second paragraph).

ii) While this paper was under review, a study, presented at the 2019 European Cystic Fibrosis Society Basic Science Conference, identified two potential binding sites of Vx-770 on the CFTR protein using in silico docking combined with site-directed mutagenesis. These findings, published in the mean time in the Journal of General Physiology (Yeh et al., 2019), have prompted us to attempt proposing and testing mechanistic kinetic models to explain the complex time courses of channel activation/deactivation that we had observed in response to addition/removal of the drug. Our modeling is indeed consistent with the presence of two independent binding sites for Vx-770 on the CFTR protein, and suggests that channel potentiation requires simultaneous binding of two drug molecules. These extensive modeling results are summarized in a new section of Results, two new figures (Figure 5 and Figure 5—figure supplement 1), and a new Table (Table 2).

iii) To clarify misunderstandings, and to better explain the physiological relevance of our findings, we now provide a more elaborate discussion of our findings. Due to the expansion of both data and discussion, the original "Results and Discussion" section has now been separated into "Results" and "Discussion".

These major additions have significantly broadened the scope of our study, which we now hope will be found suitable for publication.

Our suggestion is that you make the changes that are proposed in their appeal letter, especially in response to the minor issues raised in the previous review and that you place more emphasis on the clinical and physiological relevance of the findings.

We now discuss physiological implications at several places. In particular, we discuss the relevances (i) of protein-bound vs. free Vx-770 to drug transport vs. drug accumulation in target cell membranes (Discussion paragraph one), (ii) of aqueous solubility as an upper limit of free drug concentration in the body (Discussion paragraph one), (iii) of drug deprotonation to the distribution of the drug between the extra- and intracellular space (subsection “pH- and temperature dependence of Vx-770 solubility” paragraph one; Discussion first paragraph), (iv) of two-orders-of-magnitude higher (compared to reported values) apparent drug affinities to clinical dosage regimes (Discussion section final paragraph).

Pay attention to the suggestion of measuring solubility at 37°C and make a point of clarifying the applicability of the results to the context of biological fluids.

We have measured solubility at 37^o^C (subsection “pH- and temperature dependence of Vx-770 solubility” second paragraph), and have analyzed the thermodynamic implications of the data (second paragraph). Applicability of the results to biological fluids is clarified in the Discussion (first paragraph).

A discussion of the implication of the results for the pharmacokinetics of ivacaftor can also help to further the case for the clinical relevance of the present findings.

As mentioned above, we now discuss the relevance of protein-bound Vx-770 to drug pharmacokinetics, and of free Vx-770 to drug accumulation in target cell membranes (Discussion first paragraph). We also discuss that the measured aqueous solubility provides an upper limit of free drug concentration in the body. Finally, we discuss how deprotonation might affect the distribution of a drug between the extra- and intracellular space. We show that Vx-770 remains fully protonated (uncharged) and that, therefore, the transmembrane pH gradient and membrane potential do not affect its distribution (subsection “pH- and temperature dependence of Vx-770 solubility”; Discussion first paragraph).